# Comparative Diversity and Functional Traits of Fungal Endophytes in Response to Elevated Mineral Content in a Mangrove Ecosystem

**DOI:** 10.3390/jof9121186

**Published:** 2023-12-11

**Authors:** James Kennard S. Jacob, Katja Witzel, Thomas Edison E. dela Cruz

**Affiliations:** 1The Graduate School, University of Santo Tomas, España Blvd., Manila 1015, Philippines; jameskennard.jacob.gs@ust.edu.ph; 2Fungal Biodiversity, Ecogenomics and Systematics-Metabolomics (FBeS) Group, Research Center for the Natural and Applied Sciences, University of Santo Tomas, España Blvd., Manila 1015, Philippines; 3Department of Biological Sciences, College of Arts and Sciences, Isabela State University, Echague 3309, Isabela, Philippines; 4Department of Plant-Microbes Systems, Leibniz Institute of Vegetables and Ornamental Crops, Theodor-Echtermeyer-Weg 1, 14979 Großbeeren, Germany; 5Department of Biological Sciences, College of Science, University of Santo Tomas, España Blvd., Manila 1015, Philippines

**Keywords:** agar-based assays, anthropogenic disturbance, aquaculture practices, microbial functions, nutrient pollution, phosphate solubilization, species diversity, zinc assimilation

## Abstract

This study investigates the impact of water quality, specifically elevated phosphate and zinc content, on the diversity and functional properties of mangrove fungal endophytes in two distinct mangrove forests. Mangrove plant performance is directly related to the presence of fungal leaf endophytes as these fungi could enhance plant health, resilience, and adaptability under stressed environmental conditions. Two distinct mangrove forest sites, one non-disturbed (ND) and one disturbed by aquaculture practices (D), were assessed for differences in water quality parameters. We further analyzed the fungal endophyte diversity associated with the leaves of a target host mangrove, *Rhizophora mucronata* Lamk., with the aim to elucidate whether fungal diversity and functional traits are linked to disturbances brought about by aquaculture practices and to characterize functional traits of selected fungal isolates with respect to phosphate (PO_4_) and zinc (Zn) solubilization. Contrary to expectations, the disturbed site exhibited a higher fungal diversity, challenging assumptions about the relationship between contamination and fungal community dynamics. Water quality, as determined by nutrient and mineral levels, emerged as a crucial factor in shaping both microbial community compositions in the phyllosphere of mangroves. From both sites, we isolated 188 fungal endophytes, with the ND site hosting a higher number of isolates and a greater colonization rate. While taxonomic diversity marginally differed (ND: 28 species, D: 29 species), the Shannon (H’ = 3.19) and FAI (FA = 20.86) indices revealed a statistically significant increase in species diversity for fungal endophytes in the disturbed mangrove site as compared to the non-disturbed area (H’ = 3.10, FAI = 13.08). Our chosen mangrove fungal endophytes exhibited remarkable phosphate solubilization capabilities even at elevated concentrations, particularly those derived from the disturbed site. Despite their proficiency in solubilizing zinc across a wide range of concentrations, a significant impact on their mycelial growth was noted, underscoring a crucial aspect of their functional dynamics. Our findings revealed a nuanced trade-off between mycelial growth and enzymatic production in fungal endophytes from ostensibly less contaminated sites, highlighting the relationship between nutrient availability and microbial activities. These insights provide a foundation for understanding the impact of anthropogenic pressures, specifically nutrient pollution, on mangrove-associated fungal endophytes.

## 1. Introduction

Mangroves thrive in intertidal zones, enduring high salinity along coastal edges and brackish waters of estuaries in tropical and subtropical regions [1,2]. Evolved to withstand challenging coastal conditions, they act as a protective barrier, crucial for coastal stability. Additionally, mangroves play key roles in carbon sequestration and contribute significantly to mitigating nutrient loads from upstream sources [3]. Globally, mangrove forests span approximately eight million hectares of coastlines, with Asia hosting the majority, accounting for 42% of the total coverage [4,5]. Recognized for their dynamic nature and high productivity, mangroves are essential ecosystems [6] which play a vital role in offering essential services to coastal communities and industries, i.e., pollution control, coastal protection, timber, water filtration, and fisheries [2,7]. Notably, mangrove forests further serve as important carbon sinks, actively absorbing CO_2_ and storing carbon in both above-ground and below-ground biomass [8]. Despite the invaluable services provided by mangrove ecosystems, they have undergone significant alterations, primarily influenced by anthropogenic activities [6,9,10], monsoons, and tidal effects. The global reduction in mangrove forests has been estimated at an annual rate of 1–2%, resulting in an accumulated loss of approximately 35% since the year 2000 [7]. In Southeast Asia, aquaculture practices such as shrimp and fish farming have significantly contributed to mangrove deforestation over recent decades [11,12], leading to a reduction in the mangrove cover to 30% [7,13]. In the Philippines alone, about half of the 279,000 ha of mangrove forests was lost from 1951 to 1988 due to the establishment of fish and shrimp culture ponds [14].

The microbial community within the mangrove microbiome, encompassing bacteria, archaea, fungi, and protists, plays a crucial role in sustaining primary productivity, facilitating nutrient transformations, and contributing to vegetation conservation. Their presence is integral, with degradation activities breaking down organic matter into inorganic components. This important process allows for the absorption of nutrients into the ecosystem and the interconnected food web, highlighting the indispensable role of the microbial community that cannot be overlooked [15]. Mangrove microbial communities are actively engaged in fundamental biogeochemical cycles, particularly those governing carbon and nitrogen, which are essential for nutrient supply to both flora and fauna. Within environments marked by limited nutrients such as phosphorus and nitrogen, notably within sediments and during tidal flooding near the rhizosphere, microbes adopt specialized roles, including nitrogen fixation, phosphate solubilization, the assimilation of photosynthesis, sulfate reduction, enzyme production, and hydrocarbon degradation [16]. This interplay is profoundly shaped by the physiochemical properties of the environment and various contributing factors as described by Berg et al. [17] and Eck et al. [18].

In mangrove ecosystems, the relationship between microbial communities, nutrient dynamics, and plant responses highlights the essential interplay shaping the overall ecological dynamics. As plants are being subjected to elevated nutrient levels, they must contend with a heightened susceptibility to environmental stressors, necessitating substantial investments in root systems to enhance tolerance, particularly during conditions like drought [19]. In context, mangroves subjected to higher nutrient levels such as urea (N) and triple superphosphate (P) fertilization would cause tree mortality, indicating a discernible trend across species and biogeographic regions, highlighting the impact of eutrophication on climatic interactions in intertidal landscapes [20]. In the same manner, disturbances brought about by anthropogenic and chemical activities create a deleterious effect on other organisms including microbes. The disposal of untreated effluents from aquaculture activities can impact the diversity of microbes in natural environments, thereby influencing the health and function of aquatic ecosystems [21]. Water discharge from aquaculture shrimp farms and sewage from the developing urban areas near mangroves containing high levels of nutrients are essential contributions to pollution [22,23]. Several reports showed that shrimp-farm-impacted mangroves exhibited an elevated orthophosphate content, which increases the P form that is readily available for algae, with direct impacts on eutrophication and pollution in mangrove ecosystems [24,25,26,27]. This underlines the significance of understanding the multifaceted ecological dynamics in mangrove ecosystems, where the role of fungal endophytes adds another layer to the intricate interplay between microbial communities, nutrient cycling, and the overall health of these vital coastal habitats.

Fungi, particularly fungal endophytes, are important components of the mangrove ecosystem, playing a significant role in plant–microbe interactions [28], in nutrient cycling, and even in the bioremediation of pollutants and wastes [29]. Our present study helps fill the substantial gap regarding the influence of water quality on the diversity and functional activities of fungal endophytes associated with the leaves of *Rhizophora mucronata* Lamk. collected from two different sites in Santa Ana, Cagayan in Northern Philippines. Here, we describe the sites as non-disturbed mangrove forest (hereafter ND), intact or near-intact, and having old growth mangroves without aquaculture disturbances, including fish and shrimp pens; thus, they are less contaminated. The other site is described as disturbed (thereafter D) and is referred to as having heavily deforested mangroves and having small- to large-scale fish and shrimp pens contributing to artificially introduced nutrients. Owing to the ecological and economic importance of mangrove-associated fungi, it is important to understand how water quality influences their diversity and function. In this paper, we thus aimed to answer the question: How does water quality impact the diversity and functional traits of mangrove fungal endophytes? We hypothesized a higher fungal diversity in less contaminated sites, and their fungal functional properties remained unaffected.

## 2. Materials and Methods

### 2.1. Sampling Localities

The two sampling sites are in the municipality of Santa Ana, Cagayan, within the main island of Luzon, Philippines. The disturbed mangrove site is located in Barangay San Vicente, formerly known as Fort (18°32′44.7″ N, 122°8′54.168″ E), which bears deforested mangroves that are subjected to nutrient contaminants from adjacent aquaculture farms and household wastes. On the other hand, the non-disturbed site is located along the coasts of Palaui Island (18.5604° N, 122.1405° E). As part of the Palaui Island Protected Landscape and Seascape, the sampling area is described to have a pristine, relatively untouched mangrove condition, except perhaps from occasional visits from eco-tourists. Both sites fall under climate Type IV, where rainfall is evenly distributed throughout the year. The annual average temperature typically varies from 23 °C to 31 °C and is rarely below 20 °C or above 32 °C. The site has an annual precipitation of 2306 mm.

### 2.2. Water Quality Analyses

To determine the extent of nutrient pollution in the two sampling localities, we initially assessed the different physicochemical parameters of the water samples collected during the field sampling. The analysis followed the standard protocols for environmental water analysis by the American Public Health Association (APHA) [30]. Here, 500 mL sterile polypropylene bottles (ISOLAB Laborgeräte GmbH, Eschau, Germany, 70 × 70 × 145 mm) containing 20 mg/L sodium thiosulfate in accordance with ISO-FIS 19458/2006 for water analyses were used for water sample collection. Water samples were collected at a 50 cm depth approximately 5 m from the mangrove forest edge. A total of six (6) water samples with 500 mL volume (3 from each sampling site) were then sent to SGS Philippines to outsource standard water analyses. The following parameters were measured for this study: pH, dissolved oxygen (DO), electrical conductivity (EC), total dissolved solids (TDS). The concentrations of phosphate (PO_4_) and nitrates (NO_4_) were assessed using a spectrophotometer, while zinc (Zn) content was quantified using Atomic Absorption Spectroscopy (AAS).

### 2.3. The Host Plant and Collection of Leaf Samples

*Rhizophora mucronata* Lamk. was chosen as the source of leaves for the isolation of mangrove fungal endophytes due to the dominant presence of this mangrove species in the sampling localities. Ten healthy and mature leaves were randomly collected from different branches and twigs of the host mangrove. A total of 10 trees were sampled per sampling locality for a total of 100 leaf samples. The collected leaf samples were then pooled together in a large Ziploc plastic bag, placed in an ice box, and transported and processed in the laboratory within 24 to 48 h after collection.

### 2.4. Surface Sterilization, Isolation, and Colonization/Isolation Rates

From the pool of 100 leaf samples, 50 representative leaves were randomly chosen for each sampling locality. The mangrove leaves were initially washed in tap water to remove dirt and any attached debris. Six to seven leaf explants were cut with a one-hole puncher for each leaf sample. Following the protocol of Apurillo et al. [31] with slight modification, leaf explants with a diameter of approximately 6 mm were surface sterilized by the sequential washing of 75% EtOH (1 min), commercial bleach (5% NaOCl diluted into 1:10, 1 min), and 75% EtOH (30 s), and then washed with double-distilled water (ddH_2_O) three times. Following surface sterilization, five surface-sterilized leaf explants were plated on one petri plate containing potato dextrose agar (PDA, Carl Roth GmbH, Karlsruhe, Germany) supplemented with 500 mg/L streptomycin sulfate (Sigma-Aldrich, Burlington, MA, USA), i.e., a total of 300 explants were plated per sampling site. Streptomycin sulfate was added to inhibit the growth of leaf-associated bacteria. Tissue prints were prepared by touching leaf fragments on PDA for 10 s to test the efficacy of the surface sterilization method. The absence of any fungal or bacterial growth on the tissue prints indicates an effective surface sterilization. The plates were then incubated at room temperature (25–28 °C) and checked for fungal growth daily up to one week, and then the observation was extended up to 30 days to isolate slow-growing fungi. All fungi growing out of the leaf explants were sub-cultured on freshly prepared PDA plates for isolation. Pure cultures of the fungal isolates were maintained on PDA slants, preserved in 15 mL falcon tubes with 50% glycerol, and stored at −20 °C at the Mycology Laboratory, Research Center for the Natural and Applied Sciences, University of Santo Tomas, Manila, Philippines.

### 2.5. Diversity Assessment

#### 2.5.1. Morphospecies Identification

After successful isolation and purification of mangrove fungal endophytes, all isolates were grown on potato dextrose agar (PDA, Carl Roth GmbH, Karlsruhe, Germany) using the agar block culture technique for 5–7 days. Afterward, colony description was recorded based on colony appearance, color, form, elevation, and margin. Microscopic slides were also prepared by taking approximately 10 × 10 mm of the fungal cultures from the agar plates and placing them on clean glass slides. A drop of lactophenol blue was added and the fungal microscopic mount was observed under a compound light microscope (Leica, DM 3000, Weltzlar, Germany). Microscopic features of the fungal isolates, i.e., spore and hyphal morphologies, were described. Fungal descriptions were also compared using online databases including Facesoffungi (https://www.facesoffungi.org, accessed on 8 May 2023) and MycoBank (https://www.mycobank.org, accessed on 8 May 2023). Detailed morpho-cultural traits for each morphospecies are listed in Appendix A.

#### 2.5.2. Colonization and Isolation Rates

To assess the suitability of mangroves as hosts to fungal endophytes, we computed the percent colonization and isolation rates. Initially, the number of explants with fungal growth was counted from the 300 explants plated per sampling locality. Then, the percent colonization rate (% CR) was computed as the number of explants with fungi divided by the total number of explants multiplied by 100 [32]. The percent isolation rate (IR) was determined as the number of isolates recovered from each sampling site divided by the total number of explants multiplied by 100 [33].

#### 2.5.3. Species Accumulation Curve and Sampling Effort

To estimate the extent to which the survey was exhaustive in terms of species that were recorded in each of the study areas, a species accumulation curve was constructed based on the number of isolates recorded for each species. Species accumulation curve was successfully employed in fungal endophytes from other host plants [34,35]. We used the rarefaction formula based on the default settings of the program EstimateS (version 9.1; 100 randomizations; http://purl/oclc.org/estimates, accessed on 3 June 2023). The Chao-2 estimator was then chosen as the best estimator. The percentage of completeness for each sampling locality was then determined following the formula of Dagamac et al. [36], i.e., by dividing the actual number of recorded species by the mean number of species expected as estimated by the Chao2 estimator.

#### 2.5.4. Taxonomic Diversity

Taxonomic diversity index (TDI) was assessed by initially counting the number of species and genera and determining the ratio of the species count (S) to the genera count (G). This ratio has an inverse relationship to taxonomic diversity, meaning a lower TDI value indicates a higher level of diversity. In essence, a fungal community with species distributed across numerous genera is considered as taxonomically more diverse as compared to one where species predominantly belong to a limited set of genera, as earlier applied in the study of Dagamac et al. [36].

#### 2.5.5. Species Diversity

We assessed fungal endophyte composition and diversity between the two sampling locations by employing various species diversity indices. Species frequency, indicating the number of isolates per species, was utilized for calculating species diversity. The diversity between sampling sites was determined using the Shannon–Wiener index (H’), Simpson Index of Diversity (1-D), and Fisher-Alpha (FA) diversity statistics as used in the study of Pecundo et al. [37]. The difference in the diversity values was statistically analyzed using analysis of variance (*p* < 0.05) using Paleontological statistics software PAST (version 4.03).

#### 2.5.6. Community Analysis

To show the distribution of species between the sampling sites, a community analysis presented as a Venn diagram was created. This community analysis compares the number of shared species between the two sampling sites.

### 2.6. Assessment of Functional Traits

#### 2.6.1. Selection of Representative Fungi

Fifty-eight fungal isolates identified by morphocultural characterization as belonging to the genera *Pestalotiopsis*, *Neopestalotiopsis*, and *Pseudopestalotiopsis* were chosen for further characterization and dereplication using Matrix-Assisted Laser Desorption/Ionization-Time of Flight (MALDI-TOF) mass spectrometry-based classification of intact protein pattern. The extraction of proteins followed the ethanol-formic acid protein extraction method as also described by dela Cruz et al. [38] with slight modification on the use of metal beads for better extraction of proteins. The proteins extracts were subjected to MALDI-TOF analysis as similarly described by Djalali et al. [39]. Here, 1 µL of the fungal protein extract was initially spotted on a polished steel target (Bruker Daltonik, Bremen, Germany) and allowed to dry. Then, an equal volume of the saturated α-cyano-4-hydroxycinnamic acid (HCCA) solution was added as matrix and allowed to dry. The MALDI method was calibrated using a bacterial test standard (Bruker Daltonik, Bremen, Germany). The MALDI target plate containing the fungal protein extracts was read using an ultrafleXtreme MALDI-TOF mass spectrometer (Bruker Daltonik) working in linear positive mode and acquiring mass spectra in the range of *m*/*z* 200–20,000. Measurements were performed using flexControl v3.4 software (Bruker Daltonik). The protein spectra serve as the fingerprint of the test fungi and a dendrogram based on Principal Component Analysis (PCA) clustering of m/z values was constructed using the hierarchical method. From the cluster analysis, sixteen representative fungi, i.e., *Pestalotiopsis* (5 isolates), *Neopestalotiopsis* (5 isolates), and *Pseudopestalotiopsis* (6 isolates), from the non-disturbed mangrove forest and the mangrove forest disturbed by aquaculture practices, were chosen for the assessment of functional traits. These fungal genera are the most prevalent taxa among the isolated mangrove fungal endophytes in this study and are also known to provide beneficial properties to their host plants [40]. Detailed key morphocultural traits for these representative fungal isolates are presented in Appendix A.

#### 2.6.2. Functional Traits

Our investigation delved into two pivotal functional traits of fungal endophytes: phosphate solubilization and zinc assimilation. The phosphate solubilization assay was designed to evaluate the capacity of selected mangrove fungal endophytes in releasing phosphate (P) from tricalcium phosphate, a fundamental process influencing P-cycling and facilitating plant P uptake [41]. Conversely, the zinc solubilization assay aimed to assess the ability of these endophytes to convert zinc into an accessible form [42,43]. To mirror the diverse environmental conditions affecting mangrove hosts, we introduced three distinct concentrations of phosphate and zinc based on the values of these minerals in the water analysis into our agar-based assays: standard levels as used in the culture media (5 g/L tricalcium phosphate, 1 g/L zinc oxide), double the standard concentrations simulating nutrient-polluted mangrove environments (10 g/L tricalcium phosphate, 2 g/L zinc oxide), and half the standard concentrations replicating undisturbed mangrove areas (2.5 g/L tricalcium phosphate, 0.5 g/L zinc oxide). For the phosphate solubilization assay, 6 mm mycelial agar discs from 7-day-old fungal cultures were aseptically introduced onto chemically prepared Pikovskaya Agar (PKV, composition: glucose, 10 g/L; ammonium sulfate, 0.5 g/L; potassium chloride, 0.2 g/L; magnesium sulfate, 0.1 g/L; ferrous sulfate, 0.1 g/L; yeast extract, 0.5 g/L; agar, 15 g/L with varying concentrations of phosphates) and incubated at 25 °C. Phosphate was added as tricalcium phosphate (Sigma-Aldrich Chemie GmbH, Hamburg, Germany). Following seven days of incubation, we measured mycelial growth and halo diameter, indicative of successful phosphate solubilization. A parallel procedure was followed for the zinc solubilization assay using Basal Agar Medium (BSL, composition: glucose, 10 g/L; ammonium sulfate, 1 g/L; potassium chloride, 0.20 g/L; dipotassium hydrogen phosphate, 0.10 g/L; magnesium sulfate, 0.20 g/L; agar, 15 g/L with varying concentrations of zinc). Zinc was added as zinc oxide (Sigma-Aldrich Chemie GmbH, Hamburg, Germany). Statistical analyses were carried out, employing one-way analysis of variance (ANOVA) coupled with Tukey’s comparison test at a significance level of *p* = 0.05 to assess differences between treatments.

## 3. Results

### 3.1. Physicochemical Water Parameters

An elevated concentration of both physical (pH, dissolved oxygen, electrical conductivity) and chemical parameters (phosphate, zinc oxide, nitrates) was observed in the disturbed site as compared with results obtained from the non-disturbed site. Notably, phosphate obtained from the disturbed site was recorded six times higher than from the non-disturbed site, while zinc from the disturbed site was three times higher than from the non-disturbed. Nitrates, on the other hand, recorded a value nine times higher than that of the non-disturbed site. The results recorded from this study could be attributed to the aquaculture practices observed in the disturbed site, i.e., frequent fish feeding, prophylactic use of chemicals, and household wastewater, which led to the heightened physical and chemical parameters (Table 1).

### 3.2. Isolated Mangrove Fungal Endophytes

From the 600 leaf explants plated in this study, we isolated a total of 188 mangrove fungal endophytes. Notably, as shown in Table 2, leaf explants obtained from the non-disturbed (ND) mangrove site had a higher number of isolates (108) and greater colonization (60%) and isolation (42%) rates as compared to those leaf explants collected from the mangrove trees thriving in disturbed areas (D), with the number of isolates (60) and colonization (39%) and isolation (23%) rates. Twenty MFE (12 from ND, 8 from D) failed to produce spores and could not be identified. These were excluded in the computation of diversity values. However, the number of species between the two sampling sites did not vary significantly with the identification of 28–29 morphospecies. We added a table with detailed morphometric traits used for the identification of our fungal endophytes as a Appendix A. To assess the comprehensiveness of our sampling efforts at each site, we generated a species accumulation curve and determined the sampling effort (Figure 1). The calculated sampling effort values varied between 75% (D) and 86% (ND). This gave us insights that our sampling efforts were relatively high, though there was still a possibility of finding additional taxa as the species accumulation curve (SAC) had not yet reached a steady plateau.

### 3.3. Taxonomic and Species Diversity

From the 168 identified mangrove fungal endophytes (Table 2), the numbers of species and genera between the two mangrove sites were not relatively different, i.e., ND = 29 species, 15 genera vs. D = 28 species, 18 genera. The two mangrove sites shared sixteen species, i.e., *Cladophialophora* sp., *Cladosporium allicinum*, *C. cladosporioides*, *C. macrocarpum*, *Colletotrichum* sp., *Fusarium* sp., *F. proliferatum*, *F. solani*, *Neospestalotiopsis* sp., *N. egyptiaca*, *N. oryzae*, *P. protearum*, *Pestalotiopsis* sp. *P. glomerata*, *P. curvatispora*, and *Schizophyllum commune* (Figure 2). Comparing the taxonomic diversity index, which is basically the ratio between the number of species and number of genera, we observed a lower taxonomic diversity index (TDI) value and, hence, a high taxonomic diversity was observed in the disturbed (D) site. A similar observation was also observed when species richness and frequency of occurrence were taken into consideration. The comparison of diversity between the disturbed (D) and non-disturbed (ND) mangrove sites, based on species richness and abundance, utilized various diversity indices, including the Shannon–Weiner (H’), Simpson Index (1-D), and Fisher Alpha (FA). Notably, D exhibited higher values (H’ = 3.19; 1-D = 0.97; FA = 20.43) compared to ND (H’ = 3.10; 1-D = 0.95; FA = 13.42). These findings collectively illuminate the ecological dynamics of the two mangrove sites, emphasizing the essential role of factors beyond species richness and evenness in shaping fungal diversity and highlighting the distinct characteristics of each habitat.

### 3.4. Mineral Solubilization on Elevated Nutrient Levels

To select our target fungi for the mineral solubilization assays, we characterized all 58 isolates identified under the family Sporocadaceae belonging to genera *Pestalotiopsis*, *Neopestalotiopsis*, and *Pseudopestalotiopsis* using matrix-assisted laser desorption/ionization—time of flight (MALDI-TOF) mass spectrometry (Figure 3). The origin of the fungal isolates (D vs. ND) was plotted with the dendrogram based on the similarities of their protein profiles (Figure 3). We selected sixteen representative isolates, 5 isolates each of *Pestalotiopsis* and *Neopestalotiopsis* and 6 isolates of *Pseudopestalotiopsis.* These were distributed as follows: 10 isolates from the non-disturbed mangrove site and six isolates from the disturbed mangrove site. Detailed morphometric data for the identification of these isolates are provided in Appendix A.

The aptitude of mangrove fungal endophytes, particularly those isolated from disturbed (D) mangrove sites, to solubilize minerals and heavy metals is another interesting finding in this study. Our investigation delved into two pivotal functional traits of fungal endophytes: phosphate solubilization and zinc assimilation. At low concentrations of tricalcium phosphate, mycelial growth was observed in all 16 isolates, with a range between 9.71 and 55.20 mm in diameter after 7 days of incubation. Additionally, these mangrove fungal endophytes (MFE) demonstrated the capability to solubilize tricalcium phosphate, displaying a percentage of solubilization varying from 61.53% to 86.81% (Figure 4). Evidently, with an increase in the concentration of tricalcium phosphate in Pikovskaya Agar, there was an observed rise in mycelial growth (16.79–65.64 mm in diameter) among the mangrove fungal endophytes. However, it was noteworthy that the phosphate solubilization ability of certain MFE, particularly those derived from non-disturbed sites, appeared to be suppressed under these conditions.

In contrast to their proficiencies in phosphate solubilization, the selected MFE exhibited varying capabilities in solubilizing zinc. In culture medium with low and standard zinc oxide concentrations, these MFE consistently generated a more extensive halo zone around their fungal colonies, ranging from 23.72 mm to 74.25 mm in diameter after 5 days of incubation (Figure 5). This larger halo zone suggests an enhanced capacity for solubilizing zinc under these conditions, indicating that these MFEs can efficiently mobilize zinc in their immediate surroundings when zinc levels are at typical or lower concentrations.

Conversely, when exposed to high concentrations of zinc, a noticeable reduction in the size of the halo zone surrounding the fungal colonies was observed, ranging from 8.31 mm to 31.56 mm in diameter. This decrease in halo size implies that the MFE’s ability to solubilize zinc is influenced by the concentration of available zinc in their environment. Importantly, the growth of mycelia within the agar plates was noticeably smaller compared to the results obtained from the phosphate solubilization assay. Throughout the entire course of the assay, spanning various zinc concentrations, limited mycelial growth characterized by a small mycelial diameter was consistently observed. It is worth noting that no instances of sporulation were observed. However, despite the constrained mycelial development and the absence of spore formation, the consistent solubilizing activity toward zinc was clearly evident.

## 4. Discussion

In this study with two distinct mangrove forests, the water quality analysis showed the impacts associated with aquaculture practices (Table 1). The nutrient and mineral content (nitrate, phosphate, zinc), dissolved oxygen (DO), and total dissolved solids (TDS) are much higher in the mangrove site as affected by aquaculture practices. The notably elevated levels signify nutrient pollution, perhaps stemming from excessive fish feeding practices [44,45]. Zeng et al. [46] further noted that an alteration in the N:P ratio in water is due to the net-pen aquaculture activities. Our results concur with the study of Querijero and Mercurio [47] which showed that aquaculture and fish farming significantly impact water quality parameters in Taal Lake in Batangas, Philippines, as opposed to non-aquaculture sites. In their findings, elevated levels of pH, nitrates, and phosphates were consistently observed throughout the year, irrespective of the collection period.

We initially hypothesized that mangrove leaves collected from the non-disturbed area as characterized by better water quality with lower nutrient and mineral levels would harbor a higher number and more diverse fungal endophytes, potentially due to reduced competition and better growth conditions. Our findings showed an unexpected pattern of fungal abundance between the two sampling sites. We observed almost a similar number of species (species richness) between the two mangrove sites. However, considering species richness and abundance in computing the diversity values, we observed a higher diversity of fungal endophytes in the mangrove site that was impacted by aquaculture practices (Table 3). These findings diverged from those reported by Oono et al. [48] who observed a decrease in fungal endophyte richness in plants growing in areas with higher nitrogen–phosphorous ratios and increased sodium levels. Nutrient enrichment could increase the mortality of mangroves due to the increased vulnerability to environmental stresses [20]. We therefore believed that the mangroves in the disturbed site had a reduced resistance to environmental stresses, including perhaps a decreased tolerance to infection by invading microorganisms, thereby resulting in a higher number and more diverse fungi that colonize its leaves as endophytes, fungi which reside internally within various tissues of a host plant, existing asymptomatically and without causing any detrimental effects, and their presence may facilitate the host plant’s adaptation to both biotic and abiotic stress factors [20,49].

Mangrove forests also support a plethora of microorganisms, specifically mangrove-associated fungi. Sridhar [49] stated that mangrove fungi constitute the second-largest ecological group among the marine inhabiting fungi. These included the most common species of *Trichoderma*, *Gliocladium*, *Penicillium*, *Chaetomium*, *Mucor*, *Phyllosticta*, and *Nigrospora*. Recent studies also showed fungal endophytes associated with different plant parts of mangroves. Hamzah et al. [50] reported that the most dominant species of mangrove endophytes in the Malaysian Peninsula included *Pestalotiopsis*, *Alternaria*, *Cladosporium*, *Fusarium*, *Nigrospora*, *Phoma*, and *Xylaria*. Here, we also identified prominent genera of fungal endophytes such as *Alternaria*, *Aspergillus*, *Cladosporium*, *Colletotrichum*, *Diaporthe*, *Fusarium*, *Pestalotiopsis*, *Neopestalotiopsis*, *Pseudopestalotiopsis*, and *Phoma*. In the studies of Moron et al. [51], Apurillo et al. [31], and Ramirez et al. [52], *Alternaria*, *Aspergillus*, *Cladosporium*, *Colletotrichum*, *Fusarium*, *Guignardia*, *Nigrospora*, *Penicillium*, *Pestalotiopsis*, *Phialophora*, *Phomopsis*, and *Trichoderma* were also isolated from different mangrove plant parts—leaves, stems, and roots. These studies were conducted in the Philippines and, hence, the isolation of similar species in this study came as no surprise to us.

Our findings illuminate the crucial role of high nutrient and mineral levels in shaping the microbial community composition and diversity of mangrove fungal endophytes, driven by a combination of environmental variables. Specifically, the microbial community composition is influenced by factors associated with nutrient enrichment, including organic carbon, pH, and heavy metals primarily found within mangrove sediments which consequently affects the phyllosphere of mangroves. Differences in species richness and abundance between the disturbed (D) and non-disturbed (ND) mangrove sites were evident in various diversity indices, highlighting nuanced ecological dynamics beyond species richness and evenness. The ability of mangrove fungal endophytes, particularly those isolated from disturbed mangrove sites, to solubilize minerals and assimilate heavy metals is also another interesting finding in this study. We evaluated species of *Pestalotiopsis*, *Neopestalotiopsis*, and *Pseudopestalotiopsis* both from the non-disturbed and disturbed mangroves sites, for mineral solubilization and the assimilation of phosphate and zinc. We observed the continuous ability of the species of *Pestalotiopsis*, *Neopestalotiopsis*, and *Pseudopestalotiopsis* to solubilize zinc in basal mineral medium with low to high concentrations of zinc oxide (Figure 5). Zinc is one of the essential micronutrients required for plant growth, but Zn is mostly unavailable for assimilation by plants. Soil microbes play a role in Zn solubilization, rendering Zn as biologically available [53]. Fungal endophytes, like those isolated from the mangrove *Avicennia*, specifically *Trichoderma* sp., *Penicillium* sp., *Penicillium simplicissimum*, *Aspergillus niger*, *Abisidia spinosa*, *Abisidia glauca*, and *Abisidia cylindrospora*, can solubilize zinc [54] as similarly observed in this study. What we found interesting was the ability of our mangrove fungal endophytes to solubilize phosphate. The mangrove fungal endophytes, particularly those from the disturbed site (D), were able to effectively solubilize phosphate in all concentrations, but more importantly even at higher levels of tricalcium phosphate (added as 5–15 g/L) in the Pikovskaya Agar, while the MFE obtained from the non-disturbed (ND) site lost their ability to solubilize phosphate in an elevated level. Nutrient enrichment commonly coexists with heavy metal pollution [55]. Heavy metals frequently exert adverse effects on microbial communities, leading to reductions in microbial diversity and enzyme activities [56]. Additionally, the influence of heavy metals on the structure and function of microbial communities may surpass that of nutrient enrichment [57].

The genus *Pestalotiopsis* has frequently been documented as proficient phosphorus solubilizers under laboratory conditions [58]. We also observed a species of *Pseudopestalotiopsis* capable of solubilizing phosphate at higher levels. In fact, other reported studies like Munir et al. [58] reported the remarkable phosphorus-solubilizing abilities of *Pestalotiopsis thailandica*, an endophytic fungus isolated from Zingiberaceous species, owing to its robust solubilizing activity and the secretion of a multitude of organic acids, including phthalic acid. We believed that these fungal endophytes like those we isolated in this study are well adapted to high levels of phosphate.

Aquatic environments characterized by high contamination levels can impact not only the structural composition of fungal communities but also their metabolic functioning, as pollutants introduce additional stressors. Certain fungi adapted to such contaminated conditions may play a role in the degradation of toxic compounds, particularly in environments with lower pH, thereby facilitating the survival of other species. The observed limitation in phosphorus (P) solubilization by fungal endophytes, particularly those from the non-disturbed (ND) site, may be attributed to a trade-off described by Zheng et al. [59] between the investment in enzymatic activity and competitiveness, suggesting that fungi incapable of producing enzymes may opt for a more extensive long-distance exploration through mycelial ramification. In our study, the mangrove fungal endophytes from the ND site exhibited higher mycelial growth, potentially prioritizing exploration over enzymatic production, thus explaining their reduced ability to solubilize higher levels of P in Pikovskaya Agar media.

Furthermore, Averill et al. [60] elaborate on the notion that substrates resistant to degradation may impact the non-release of enzymatic activities from fungal cultures. This phenomenon could influence carbon turnover, leading to a lack of enzymatic activity for P solubilization. In such cases, the increased release of carbon onto substrates may result in a higher mycelial growth rate. Hence, our findings align with the literature, suggesting that the reduced ability of ND fungal endophytes to solubilize higher levels of P may be linked to their preference for mycelial growth over enzymatic production in response to substrate characteristics, supporting the trade-off concept proposed by Zheng et al. [59]

## 5. Conclusions

In conclusion, our investigation sought to address the question: “How does water quality impact the diversity and functions of mangrove fungal endophytes”? Contrary to our initial expectation, this study revealed an unexpected pattern of fungal abundance between a contaminated mangrove site affected by aquaculture practices and a less contaminated, non-disturbed site. Although both sites exhibited a similar number of fungal species (species richness), the disturbed mangrove site subjected to aquaculture practices displayed a higher diversity of fungal endophytes when considering both species richness and abundance. Furthermore, our findings indicate that water quality, influenced by nutrient and mineral levels, plays a pivotal role in shaping the fungal community composition and diversity of mangrove fungal endophytes, as well as in affecting microbial functions within these ecosystems underpinning their functional significance in nutrient cycling processes. Our findings underscore the complexity of interactions within mangrove ecosystems and provide valuable insights for future research on the plant–microbe–environment interaction, environmental management, and conservation efforts in these critical coastal habitats.

## Figures and Tables

**Figure 1 jof-09-01186-f001:**
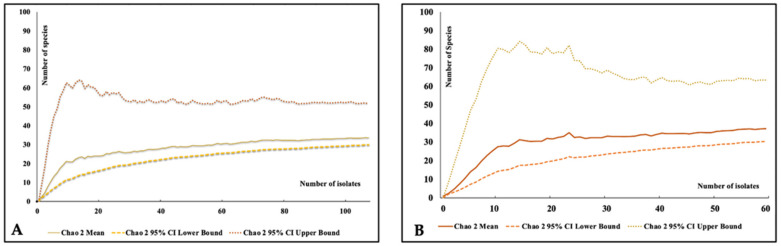
Comparative analysis of Chao2 estimates between species richness patterns in (**A**) disturbed (D) and (**B**) non-disturbed (ND) mangrove ecosystem.

**Figure 2 jof-09-01186-f002:**
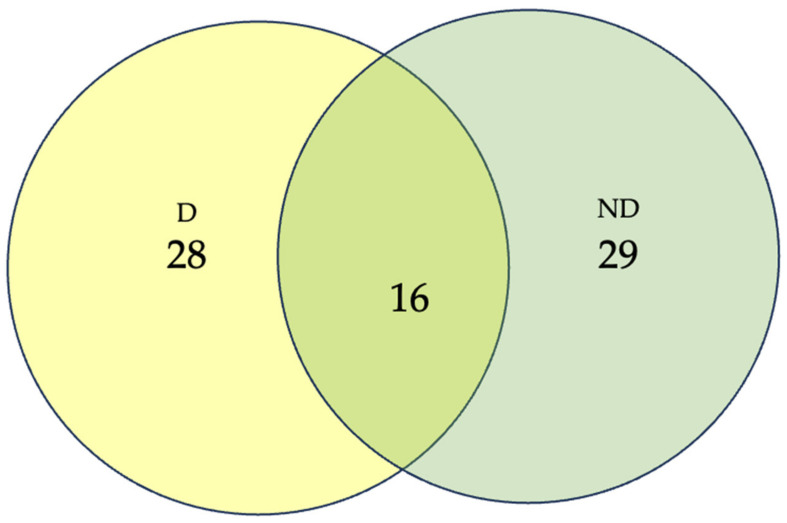
Venn diagram showing the distribution of the identified mangrove fungal species between the two sampling sites.

**Figure 3 jof-09-01186-f003:**
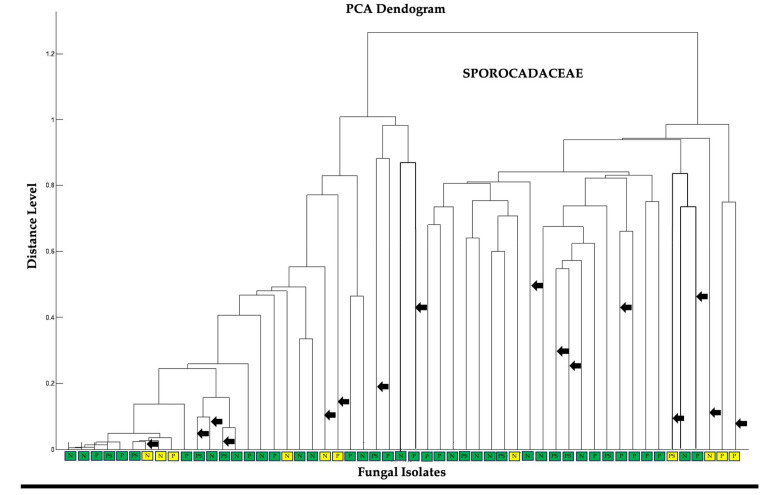
Principal Component Analysis (PCA) dendrogram illustrating the taxonomic relationships of 16 representative mangrove fungal endophytes from the family Sporocadaceae, including genera *Pestalotiopsis* (P), *Neopestalotiopsis* (N), and *Pseudopestalotiopsis* (PS). Fungal species isolated from the non-disturbed (ND) site are highlighted with a green box, while those from disturbed site (D) are represented by a yellow box. Black arrowheads indicate the positioning of the selected 16 mangrove fungal endophytes in the dendrogram.

**Figure 4 jof-09-01186-f004:**
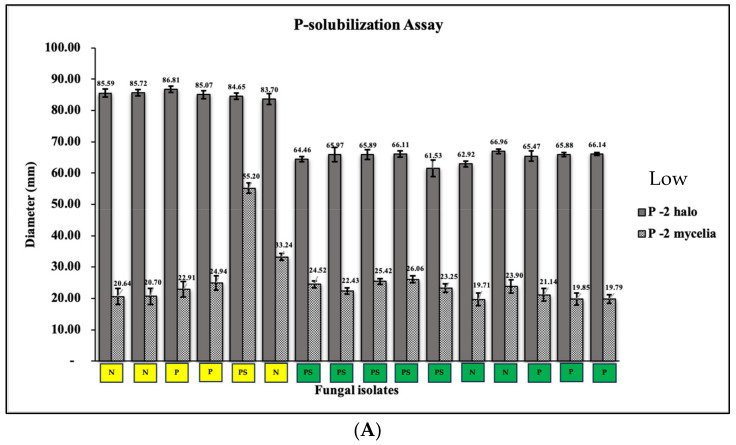
Solubilization of phosphate (P) by the selected mangrove fungal endophytes in Pikovskaya Agar medium supplemented with (**A**) low, (**B**) standard, and (**C**) high concentrations of tricalcium phosphate. Fungal species isolated from the non-disturbed (ND) site are highlighted with a green box, while those from disturbed site (D) are represented by a yellow box.

**Figure 5 jof-09-01186-f005:**
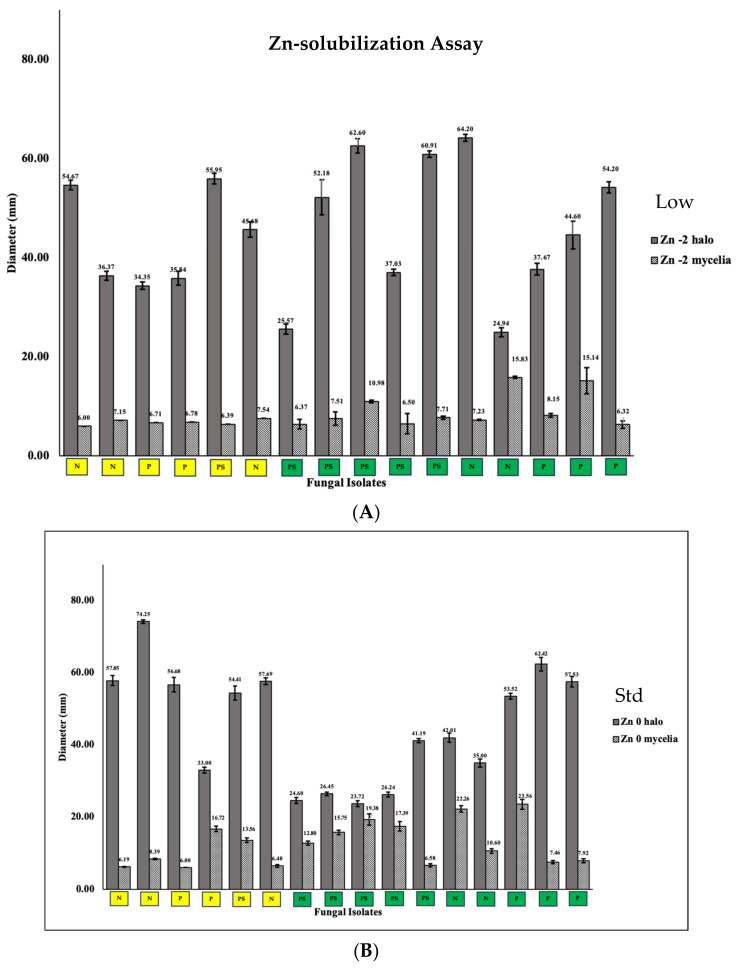
Solubilization of zinc (Zn) by the selected mangrove fungal endophytes in modified basal medium supplemented with (**A**) low, (**B**) standard, and (**C**) high concentrations of zinc oxide. Fungal species isolated from the non-disturbed (ND) site are highlighted with a green box, while those from disturbed site (D) are represented by a yellow box.

**Table 1 jof-09-01186-t001:** Comparative water physicochemical analysis in two mangrove sampling localities.

Sites	Chemical Parameters			Physical Parameters			
	Phosphate	Zinc	Nitrates	pH	Dissolved Oxygen (DO)	Electrical Conductivity (EC)	Total Dissolved Solids
D	26.40 ± 0.00 ^a^	15.41 ± 0.01 ^a^	78.84 ± 0.03 ^a^	8.30 ± 0.00 ^a^	8.00 ± 0.01 ^a^	1984.50 ± 0.27 ^a^	2461.00 ± 1.00 ^a^
ND	4.30 ± 0.00 ^b^	5.12 ± 0.00 ^b^	8.02 ± 0.05 ^b^	7.40 ± 0.03 ^b^	20.00 ± 0.06 ^b^	850.51 ± 0.32 ^b^	915.00 ± 1.00 ^b^

Note: presented are means ± SD; values followed by the same letters are not significantly different, *p* < 0.05.

**Table 2 jof-09-01186-t002:** Occurrence and diversity of mangrove fungal endophytes in the two sampling localities.

Morphospecies	^1^ Sample Source	
	D	ND
*Alternaria alternata* (Fr.) Karst	1	0
*Aspergillus parasiticus* Speare	0	1
*Aspergillus* sp.	1	0
*Aspergillus terreus* Thom	4	0
*Cladophialophora bantiana* de Hoog, Kwon-Chung & McGinnis	0	1
*Cladophialophora* sp.	3	12
*Cladosporium allicinum* (Fr.) Bensch, U. Braun & Crous	1	1
*Cladosporium cladosporioides* de Vries	1	5
*Cladosporium herbarum* (Pers.) Link	0	1
*Cladosporium macrocarpum* Preuss	1	1
*Cladosporium* sp.	0	1
*Colletotrichum* sp.	1	3
*Diaporthe* sp.	0	9
*Exophiala* sp.	1	0
*Fusarium* sp.	2	3
*Fusarium proliferatum* Gerlack & Nirenberg	4	5
*Fusarium solani* Sacc.	3	4
*Leptosphaeria* sp.	1	0
*Lophiostoma* sp.	1	0
*Neopestalotiopsis clavispora* Maharachch, Hyde & Crous	0	5
*Neopestalotiopsis egyptiaca* Ismail, Perrone & Crous	2	1
*Neopestalotiopsis rhizophorae* Norphanphoun, Wen & Hyde	2	2
*Neospestalotiopsis* sp.	4	7
*Nigrospora oryzae* (Berk. & Broome) Petch	3	2
*Penicillium oxalicum* Currie & Thom	0	2
*Pestalotiopsis maculans* (Corda) Nag Raj	0	2
*Pestalotiopsis microspora* (Speg.) Zhao & Nan Li	0	5
*Pestalotiopsis protearum* Maharachch, Hyde & Crous	2	3
*Pestalotiopsis* sp.	3	6
*Phaeosphaeriopsis* sp	0	1
*Phoma glomerata* Chen & Cai	4	2
*Phoma* sp.	4	0
*Phyllosticta* sp.	2	0
*Pseudopestalotiopsis curvatispora* Norphanphoun, Wen & Hyde	1	4
*Pseudopestalotiopsis* sp.	0	9
*Ramichloridium biverticilliatum* Arzanlou & Crous	0	3
*Schizophyllum commune* Fr.	3	7
*Talaromyces* sp.	1	0
*Ulocladium chartarum* Preuss	2	0
*Ulocladium* sp.	2	0

^1^ Sample source: D = disturbed, ND = non-disturbed.

**Table 3 jof-09-01186-t003:** Records and diversity of mangrove fungal endophytes associated with the leaves of *Rhizophora mucronata*.

Sample Source	^1^ Records and Diversity Indices								
	R	G	S	TDI	H’	FA	1-D	CR (%)	IR (%)
disturbed (D)	60	18	28	1.56 ^a^	3.19 ^a^	20.43 ^a^	0.97 ^a^	39	23
non-disturbed (ND)	108	15	29	1.93 ^b^	3.10 ^b^	13.42 ^b^	0.95 ^b^	23	40

**^1^** Records and diversity indices: R = individuals/records, G = number of genera, S = number of species, H’ = Shannon diversity index, FA = Fisher’s Alpha diversity index, 1-D = Shannon–Weiner diversity index, CR = colonization rate, IR = isolation rate. Values followed by the same letter are not significantly different, *p* < 0.05.

## Data Availability

Data is contained within the article and Appendix A.

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
