# Peer review of "Comparative Diversity and Functional Traits of Fungal Endophytes in Response to Elevated Mineral Content in a Mangrove Ecosystem"

_jof, 2023, doi:10.3390/jof9121186_

Round 1

Reviewer 1 Report (Previous Reviewer 1)

Comments and Suggestions for Authors

This revised version of the manuscript is improved and addressed the concerns that I had in the first version. The results are interesting and important. 

Comments on the Quality of English Language

While the manuscript is easily understandable, it would be improved with some editing.  This is not essential. 

Reviewer 2 Report (Previous Reviewer 2)

Comments and Suggestions for Authors

The authors made considerable improvements to the manuscript. I believe it is now ready for publication.

This manuscript is a resubmission of an earlier submission. The following is a list of the peer review reports and author responses from that submission.

Round 1

Reviewer 1 Report

Comments and Suggestions for Authors

While there is high recognition of the importance of mangrove ecosystems there has been a paucity of studies on the microbial community, including fungi.  This gap in knowledge limits appropriate management and policy decisions to mitigate loss / degradation of these vital ecosystems.  So, this study of endophytes in disturbed and undisturbed sites is needed. The paper is generally well written and easy to follow, and methods and analyses are appropriate.  However, I had a couple of questions about the interpretation of the results. 

·      Much is made that the trees in the undisturbed site are larger than in the disturbed site.  But they are also nearly 3x older (3.85 vs 10.30). How much of the documented size differences are due to disturbance versus age? 

·      I found lines 510-514 confusing. I did not see data to support the statement that the disturbed sites had reduced forest complexity or a shift in dominant tree species composition.  Are there data that can be added to support the statement?

·      The authors report that endophytes from the undisturbed site were able to solubilize P higher concentration of P than those in the disturbed site.  However, the undisturbed site had much lower concentration of PO4 than the disturbed site. Perhaps the authors can add something about the potential ecological significance of this.

·      I did not understand the discussion about tradeoff between enzyme production and mycelial growth in lines 571-580.  This needs to be rewritten to improve clarity.   

The data in this paper are important.  Addressing the points above will clarify and strengthen the implications of the results and increase the papers impact. 

Comments on the Quality of English Language

For the most part, the quality of English is fine and the paper is easy to read.  But, please note the couple of sections I call out in my review above.  My questions may be due to not understanding the text.

Reviewer 2 Report

Comments and Suggestions for Authors

This work is very interesting and relevant. I strongly believe in the potential of the results, but I have no way of validating the identifications of the isolates, as the technique of morphological studies for isolates in petri dishes is not sufficient to guarantee identification at a specific level. I believe that in these cases only the analysis of nuclear and ribosomal regions can guarantee the validity of the identifications. As I think that in the current state the authors will not carry out analyzes of this type, I suggest rewriting the entire manuscript to give a sense of an approximate determination of taxons in the order of operational taxonomic units and not of species. This will greatly influence the discussions and, in particular, in terms of statistics regarding wealth and diversity, but we cannot leave these determinations as certain as they cast doubt on all the results presented. If the authors can still do so, I suggest that molecular analysis be carried out to certify and validate the identification determinations.

Reviewer 3 Report

Comments and Suggestions for Authors

Please find the comments and suggestions below:

Comparative diversity and functional traits of fungal end0phytes isolated from non-disturbed and aquaculture-disturbed mangrove ecosystems.

Comments and suggestions

Abstract

Lines 22-24. Very broad only descriptive but not precise scientific

Give a definition for DPA. Also, it needs to be supported by what kind(s) of practices.

Linea 34, the description “higher value” very vague and not scientifically precise.

Lines 37-43. Very broad with no precise layout to any of significant results.

Keyword

“agar-based assays”, “disturbance”,” microbial functions”, “species diversity”, and “tropical fungi” Basically all are not sound useful keywords and do not directly relate a specific lietrature. Also, it is unusual to use abbreviations and/ or use acronyms in keywords.

Introduction

Lines 48-56. This is very general for all kinds of water bodies and needs to be specific to the mangrove.

Lines 57-65. Also, it is very general and does not directly relate to the present situation and this research work.

Lines up to 75. Also, it is just jargon in general and not presenting a specific observation that became a phenomenon which worth to be hypothesize about and then scientifically experimented and disapprove the nil hypothesis.

Lines 77-84. This needs to be dealt with in accordance with the basics of the scientific resaerch outlines in the point above. It needs to be turned around presenting observation(s), that became phenomenon(na) and then laying out a hypothesis and the scientific experimentation. Was there any of this observed and became a phenomenon(na) at the site(s) of this research, “Destruction of mangrove ecosystems can decrease the number of substrates available for colonization by fungi, thereby reducing species richness of mangrove-associated fungi”. What destruction occurred to and/or at the researched site(s)???

For the rest of the introduction lines 85-96, it is still not satisfying the basic justification and putting this research in its suitable significance to be dealt with and further investigated.

Materials and methods

Line 101. “Pronounced maximum rain” very ambiguous. Rain fall has numerical values. Also, “Dry’ needs to be more specific by numerical rain fall vales.

Lines 108-111. This is a very vague general description of what ought to be the main justification of considering this study. For instance, saying “extremely modified” is not an exact scientific evaluation to justify and be fit to do technical scientific study(ies). Furthermore, this should be well illustrated in Figure 1.

Figure 1. This adds no positive value to justify and adjust carry out this work. The picture (color photo) shows the ND situation, while it is more needed to show the DAP. This figure needs to be reconsidered and modified to show the extent of DAP. There is no added value for having a large map of the islands. The small enlargement on the upper right corner could show more details. More and closer pictures of the damage would be essential.

As an example, note that the map published in Ref. 15 is more informative to that site than what is presented here for this study.

Water sampling

“Water samples were collected at 50-cm depth approximately 5 meters from the mangrove forest edge”. How long along that water shore or the edge of the forest that water samples were collected?

Lines 145-155. The description is not in agreement with cited reference number 15. It is more exact to give the GPS data for the line(s) of sampling rather than only the data for the locations in general on the mainland and the island. Also, note in that reference, there were FOUR transect lines.

“Data Collection and Analyses

Four line transects of variable lengths (200-2000m), depending on forest extent, were laid from the landward to seaward margin at each of the four sites (Village undisturbed and disturbed and Swere undisturbed and disturbed).” This is a quotation from Ref 15.

The present ONE transection line is not enough to cover sampling for a seashore location so many meters (??m) at the affected location and way up into the vegetation away from that shoreline. So, the question is, how long are those shorelines?

Lines 148-149. “Tree traits were recorded at intervals of 3 - 5 m along the 50 m transect line. A total of 30 trees were selected and measured for each sampling site” It is not clear. The 50 m long transect line divided by average 3-5m interval give 8-9 trees checked. Now how did the number of 30 trees was reached? And, how the data in Table 1 generated from those measurements and presented. Are those readings averages of how many background readings. Where is the statistical result to this data. 

Line 157. “Without any visible signs of diseases”. Please note that “signs” refer to the pathogen and symptoms refer to the disease.

Lines 159-160. Those trees are along a 50m transection line. The question is that one line enough to represent that large location?   

Lines 206-216. This is a very classical, old school, way of identifying fungal isolates. It needs to be supported by pictures of the isolate showing macro and micro morphology. However, it would have been more exact and confident to be identified using the PCR technology.

Lines 224-227. The following phrasing of “While taxonomic diversity focuses on species richness, we also used different species diversity indices to compare the fungal endophyte composition and diversity between the two sampling localities. Species frequency refers here to the number of isolates per species. These data as occurrence were used for the computation of species diversity.” Needs to be reconsidered.

It needs to be rewritten in a simple direct statement.

The statical differences expressions needs to be obviously presented.

Line 239. “Matrix-Assisted Laser Desorption Ionization Time-Of-Flight mass spectrometry (MALDI-TOF MS).” Needs to be presented clear full name(s) preceding the acronym as it was in the reference.

Also, in line 246. “HCCA (α-cyano-4-hydroxycinnamic acid solution)”

Note: Discrepancy in author names reference cited as Djalali Farahani-Kofoet et al. [38], but it is

38. Djalali, F.K.; Witzel, K.; Graefe, J.; Grosch, R.; Zrenner, R.

Line 253. Full name of PCA required.

Lines 254-255. “From the cluster analysis, sixteen representative fungi, i.e., Pestalotiopsis (5 isolates), Neopestaloitiopsis (5 isolates), and Pseudopestalotiopsis (6 isolates)this is not clear, and it is not supported by what is presented in Table 2.

Lines 259-276. It could be derived from statement in these line that fungi (endophytes) isolated from leaves, artificially grown on agar media, used on separate specially prepared another artificial agar media and checked for their invitro biochemical activity. Then in what way and how that (those ) fungi affect the sea water away??? And/or inflict such chemical changes.

Results

Lines 281-287, and Table 1. There are no statistical analysis shown.

Table 1. It is unusual using mixed measurement values, inches and centigrade values in the same table. It needs to be of one kind.

Also, Table 1 and all other tables, graphs and pictures, the title, caption needs to be descriptive to the extent that it makes that presentation easily and fully comprehended as it stands alone. In this case the title should be rewritten.

Also, note in the absence of background measurements and results of statistical analyses and probably pictures to illustrate the general and closer observation. Unfortunately the data presented in this table sounds bias rather that factual.

Lines 297-298. The statement; “Interestingly, twenty fungal isolates (7 from ND, 13 from DAP) did not produce spores even after an extended incubation period.” That is a bold statement. Mycologically, several of the Heltiales endophytes are non-spore producers. Besides, statement in lines 299-301 in not a mycologically sound statement. It needs to be deleted. Also, it reveals a very significant negative default to this work of ignoring the most dependable and recognized technology, the PCR molecular termination of the isolated fungi.

Line 304. species accumulation curve (SAC).

Lines 291- 309 including Figure 2. It needs further clarification.

-          Values, percentages where did they come from, where are they shown??

-          The statement: “To gauge the completeness of our sampling in each locality, we constructed a species 303 accumulation curve and computed the sampling effort (Fig. 2). The computed sampling effort values ranged from 75% (DAP) to 86% (ND).” Not clear ??

-          Figure 2. Caption needs to be rewritten showing what Chao-Mass and… stand for.

-          Is it number of species or it is number of isolates?? Many Is fungal isolates could be obtained but it does not mean all are species. It is here also sounding a bias presentation.

Table 2. shows total isolates from both sites is 168. However, it was mentioned earlier that total isolates were 800 on line 209!!

Lines 347-359 including Figure 4.

-          Needs further clarification. The present selection was based on the structural protein print of those fungi. And further postulated to correlates to the fungal extracellular activity(ies) when those fungi are alive? This sound of no sound scientific and/or significance. Needs to be reconsidered all over.

-          Figure 4. The caption needs to be rewritten. The content is not clear. What does the tiny alphabet in the small boxes stand for. Also, the Black Arrow heads.

Similar comments to Figures 5 and 6 as presented above for other figures.

Also, as mentioned above, those are results of fungal isolates living away from their original natural environment and living under completely artificial conditions. So, it is hardly seen how it would correlate to changes at away environmental conditions under a very vast variable dynamics of the seahorse condition. There is a serious missing link in the line of thinking and work.

Discussion

Lines 487-497 and similar, are a spillover from the introduction. Needs to be deleted.

Note, that the main core of the published literature below is focused upon the significant factors in such habitat disturbance. However, the present work completely ignored those factors.

“The notably elevated levels signify nutrient pollution, perhaps stemming from excessive fish feeding practices [44,45]. Zeng et al. [46] further noted that an alteration in the N:P ratio in water is due to the net-pen aquaculture activities.”

Note, Ref 46 dealt with live bacterial communities.

Lines 548-581. This is a very far and not well propounded extrapolation. Needs to be reconsidered. The endophytes study and its finding can be a stand-alone for the vegetation of any habitat within the any inflicted change in vegetation, environment and / or any adverse or conducive practices(es). And it was shown here that there is no significant discrepancy in this respect.

Comments on the Quality of English Language

Good English language